# Volatility and Herding Bias on ESG Leaders' Portfolios Performance

**Nektarios Gavrilakis and Christos Floros \***

Department of Accounting and Finance, Hellenic Mediterranean University, 71410 Heraklion, Crete, Greece;
ngavrilakis@hmu.gr
\* Correspondence: cfloros@hmu.gr

**Abstract:** We here analyze the factor loadings given by the CAPM, the Fama–French three (FF3), and the five-factor model (FF5), and test the performance and the validity of adding two more factors (volatility and dispersion of returns) to the FF5 factor model of European index-based ESG leaders' portfolios. Our ESG leaders' portfolios generated significant negative alphas during 2012–2022, corroborating the literature's negative argument. The negative abnormal returns of ESG leaders' portfolios are homogeneous across the three ESG pillars. We conclude that European ESG leaders' portfolios are biased toward large cap and value stocks with robust operating profitability and against aggressive investments. As robustness tests, we examine Global ESG leaders' index-based portfolios, producing the same results but with reduced importance in some loading factors like profitability and investment strategy. Furthermore, we deduced that European and Global ESG leaders' portfolios tilt towards volatility and herding bias.

**Keywords:** ESG; Jensen's alpha; volatility; herding

## 1. Introduction

The Environmental Social and Governance (ESG) market is expected to have increased its assets under management (AuM) to USD 33.9 tn by 2026, from USD 18.4 tn in 2021 (Ross 2023). PricewaterhouseCoopers (2022) assumes ESG investing will be a potential opportunity in the coming years. European investors develop increased demand for ESG products despite headwinds due to rising interest rates, constant high inflation, and recession fears (Morningstar Manager Research 2023). To advance the effort toward ESG investing in the sustainable financial sector, it is essential to understand the effects of ESG leaders on portfolio performance.

Our study contributes to the relatively limited research on sustainable investing, and addresses whether investing in European ESG leaders could yield positive risk-adjusted returns. Beyond measuring Jensen's alpha against the market benchmark and the factor loadings given by the Capital Asset Pricing Model (Treynor 1961, 1962), the Fama and French (1993) three-factor model (FF3) and the Fama and French (2015) five-factor model (FF5), we analyze the performance and the validity of adding two more factors (volatility and dispersion of returns) to the FF5 model. In this way, we might address two more issues: if ESG leaders' portfolios are biased toward volatility and dispersion of returns and if ESG leaders' portfolios lean toward herding.

As proxies for ESG leaders, firstly, we selected a sample consisting of European ESG leaders' indices (Stoxx Europe ESG Environmental Leaders, Stoxx Europe ESG Social Leaders, Stoxx Europe ESG Governance Leaders, Stoxx Europe ESG Leaders select 30), and as a benchmark, we selected the Stoxx Europe 600 Index. Secondly, for the robustness check, we selected a sample consisting of Global ESG leaders' indices (Stoxx Global ESG Environmental Leaders, Stoxx Global ESG Social Leaders, Stoxx Global ESG Governance Leaders, Stoxx Global ESG Leaders), and as a benchmark, we selected the Stoxx Global 1800 index. Data were obtained from the Qontigo (2023) database, which uses ESG scores in environmental, social, and governance pillars from the Morningstar/Sustainalytics

(2023) provider "https://www.sustainalytics.com/esg-data (accessed on 7 July 2023)". The limited availability of data on EGS index daily prices forced us to use the period from 1 June 2012 to 14 July 2022 from the above provider. de Oliveira et al. (2020) argued that ESG indices are a valuable investors' tool that can be used to reduce the fear of increased uncertainty.

The literature on ESG and portfolio performance is developing, motivating us to focus on this area. Some studies proposed ESG integration as a tool for portfolio optimization (Henriksson et al. 2019) and confirm superior returns (Renneboog et al. 2008; Consolandi et al. 2009), while others (Bauer et al. 2005; Adler and Kritzman 2008; and Berlinger and Lovas 2015) show evidence of underperformance in ESG portfolios. Studies by Managi et al. (2012) and Hartzmark and Sussman (2019) support neutrality, meaning that ESG markets are efficient, and managing positive risk-adjusted returns is impossible. To help address the effect of ESG tilting on ESG leaders' performance, we examine the factor loadings given by the CAPM, the FF3, and the FF5 models, and provide some practical implications for corporate decision-making and portfolio construction processes.

Our second motivation is to identify the validity of adding two more factors (volatility and dispersion of returns) to the FF5 factor model. Volatility behavior is a dominant factor when selecting portfolio construction and performance assets. Like the Chicago Board Options Exchange (CBOE) Volatility Index (VIX), a volatility index may capture short-term mood sentiments and is an effective proxy instrument in behavioral finance. The VIX index displays the volatility expectations over the next 30 calendar days (Marquit and Curry 2023) and is considered a real-time monitoring index. As Siriopoulos and Fassas (2009) explain, the CBOE Volatility Index has been broadly accepted due to its advanced interpretative power in antithesis to historical volatility. Investors, fund managers, and traders use the VIX as a barometer to determine the market's fear or level of risk when constructing their portfolios (Economou et al. 2018). The current study uses the VIX index based on the CBOE VIX methodology (Whaley 2000) to capture any ESG leaders' investors' bias in relation to volatility. Furthermore, the present study examines whether European and Global ESG leaders' portfolios tilt towards the dispersion of returns by employing the cross-sectional absolute deviation of returns (CSAD), introduced by Chang et al. (2000), which is the most accepted method of return dispersion used in behavioral finance literature. In this way, we fill the gap in the literature and investigate any bias in the dispersion of returns on ESG leaders' investments.

An interesting avenue of research arising from the above discussion will be to examine if there is any herding behavior in ESG leaders' portfolios, motivating us to extend the behavioral finance literature. Herding behavior refers to imitating and following others' investment decisions instead of following one's beliefs and information. In the investment process, emotional biases can arise due to herding affecting investment decisions in selecting, buying, holding, or selling assets. Understanding and analyzing herding behavior is vital to the portfolio strategies of investors and fund managers. The cross-sectional absolute deviation of returns (CSAD) is the most common measure used to examine herding behavior, introduced by Chang et al. (2000). There is a small volume of existing studies investigating herding behavior using ESG data. Rubbaniy et al. (2021), Blondel (2022) and Gavrilakis and Floros (2023a) found evidence of herding behavior using ESG data, while Ciciretti et al. (2021) found anti-herding behavior.

The present study reports negative alphas on European ESG leaders' portfolios by using the CAPM, FF3, and FF5 regression models. The negative abnormal returns of ESG leaders' portfolios are homogeneous across the three ESG disclosures (environmental, social, and governance). Moreover, the evidence suggests that European ESG investing tilts towards large caps, and values stocks with robust operating profitability against aggressive strategies. We examine Global ESG leaders' index-based portfolios in a robustness check, reporting the same outcomes but with insignificant importance in some loading factors like profitability and investment strategy. Furthermore, we document for the first time that ESG leader's portfolios are biased toward volatility and herding behavior. These

results shed light on risk–return tradeoffs and behavioral biases on the part of investors and finance professionals responding proactively and reducing uncertainty in the portfolio optimization process.

The current study contributes to the literature by providing valuable updates for investors and fund managers exposed to ESG leaders' assets. We show investors and professionals cannot achieve high-risk-adjusted returns when holding ESG leaders' stocks in the short run. This outcome is helpful during portfolio construction and diversification processes. The evidence that European ESG investing tilts towards large caps, and values stocks with robust operating profitability against aggressive strategies encourages their use in corporate financial decisions. In addition, understanding and exploring how the volatility and dispersion of returns interact with ESG investing is essential for constructing optimal portfolios. Finally, our work contributes by reporting for the first time that ESG leaders' portfolios are biased toward volatility and herding behavior, which is helpful for investors, fund managers, and analysts seeking to reduce risk-taking and hedge their portfolios.

The rest of this study is organized as follows: In Section 2, we review the literature and analyze alphas, volatility, and herding. In Section 3, the methodology is presented, while our results are reported in Section 4. Finally, Section 5 concludes the study.

## 2. Literature Review

### 2.1. ESG Abnormal Returns

Literature on ESG investment performance focuses on three main areas. The ESG investments result in positive returns, as Renneboog et al. (2008), Consolandi et al. (2009), Giese et al. (2019) and Yu (2022) claimed. Secondly, the ESG investments underperform, mainly due to lower diversification and short-term growth immolation, as supported by Hamilton et al. (1993), Carhart (1997), Bauer et al. (2005), Adler and Kritzman (2008) and Berlinger and Lovas (2015). Lastly, the neutrality argument is supported by Managi et al. (2012), Mollet and Ziegler (2014), Hartzmark and Sussman (2019), and Naffa and Fain (2022), who report insignificant abnormal returns, supporting the efficient market hypothesis (EMH) that all existing market information is reflected in asset prices, and alpha generation is impossible.

Following the work of Merton (1987), Heinkel et al. (2001), Zerbib (2020), and Pedersen et al. (2021), we presume that investors sensitive to ESG are averse to the assets of laggard ESG firms, and place increased demand, especially on the part of younger generations (Tao et al. 2020), on ESG leaders. Financial market participants have recently reported obligations (article 8, 9) to promote assets with high rather than low ESG scores (Chava 2014), and intentionally incorporate ESG leaders into their sustainable investment portfolios (Tao et al. 2020). In measuring Jensen's alpha abnormal returns, Zehir and Aybars (2020), using the CAPM regression model, reported that two ESG-based portfolio scores underestimate the market index, while, when using the FF3 model, the abnormal returns of the ESG bottom governance portfolio perform better than the rest of the market. In the same way, Teti et al. (2023), by employing the CAPM, FF3, and FF5 methodologies, found robust evidence that a bottom decile portfolio produces negative alphas. In a recent paper, Luo (2022) examined UK stocks from 2003 to 2020, and found that companies in the low ESG quintile perform better (value-weighted returns) by 0.513% per month than those in the high ESG quintile. In addition, Luo (2022) analyzed the returns of ESG portfolios constructed from stocks of the STOXX Europe 600 index, and found that the environmental, social, and governance premiums are significant. In their study, Naffa and Fain (2022) measured the performance of Global ESG assets by recommending a new FF5 factor approach, and deduced that ESG portfolios did not yield significant alphas during 2015–2019, verifying the neutrality argument of the literature. Similarly, Dhasmana et al. (2023) investigated the relationship between the MSCI ESG index and investor sentiment, and revealed the underperformance of the MSCI ESG index in India. Following these leads, we provide a route to the first hypothesis:

**H1.** *ESG leaders' portfolios produce negative alphas.*

### 2.2. ESG, Volatility, and Dispersion of Returns

Financial assets present volatility clustering and asymmetric behavior through financial uncertainty. Various volatility methods have been adopted to analyze the volatility behavior of asset returns (McKenzie and Mitchell 2002). The literature review here indicates some studies that have concentrated on the use of the volatility index (VIX) as a proxy for investor sentiment, with mixed results. The VIX index is known as the "investor fear gauge", and captures investors' expectations of market volatility (Durand et al. 2011).

In their paper, Durand et al. (2011) analyzed the role of the VIX and momentum in asset pricing, and concluded that the explanatory power of all factors in Fama and French's three-factor model is enhanced by the inclusion of both the momentum and VIX factors. Chiang (2012), by using the GARCH methodology, examined the relationship among the S&P500, NASDAQ100, VIX, and VXN indices, and concluded that the VIX—fear index significantly affects the S&P500 Index. In the same way, López-Cabarcos et al. (2019) examined the effects of the VIX index on the indices of the S&P500 and the S&P500 environmental and social responsibility from 2015 to 2016, and found a negative correlation. Morales et al. (2019) reported a negative effect of the VIX index on Socially Responsible Investing (SRI) indices (DJSI World excluding alcohol, gambling, tobacco and arms index, Dow Jones Sustainability Index, MSCI KLD 400 Social index, and World USA subset index). Chen and Gao (2020) examined how three defined volatility risk factors derived from VIX may affect the pricing of assets by employing the three-factor Fama–French asset pricing model, and identified a significant relation of the risk factors to returns of individual portfolios. In their study, Öcal and Kamil (2021), using as risk indicators the VIX, the CDS, and FX volatility indices, revealed that stocks with higher ESG exposure in Germany, France, Indonesia, and Turkey are less affected by market crises than companies included in broad-based indices. Furthermore, Shaikh (2022) examined the relationship between sustainable investment and uncertainties, and found a negative relationship between the VIX index and Dow Jones Sustainability Indices from 2000 to 2017. Vergili and Celik (2023), in their study, reported a long-term cointegration relationship (with a negative coefficient) between the VIX fear index and the Dow Jones Sustainability Emerging Markets Index (DJSEMUP) by using monthly data from 2013 to 2020. Finally, by employing the Morgan Stanley Capital International (MSCI) ESG Leader index, Sabbaghi (2023) examined the volatility risk and provided seminal evidence of a slow response to news in emerging markets on the part of the selected index. A relatively new index, the EURO STOXX 50 Volatility Index (VSTOXX), provides the implied volatility over the next 30 days on the EURO STOXX 50 Index. Stanescu and Tunaru (2013), by adding VIX and VSTOXX volatility index futures to a portfolio, improved the return–risk profiles of portfolios, particularly during turbulent times. Cocozza et al. (2021) found the same behaviors in the VSTOXX and VIX indexes in their study.

According to Gleason et al. (2004) and Henker et al. (2004), dispersion measures the extent to which investors follow the market's expectations. Ramadan (2015) stated that "if investors follow market expectations, then their returns will not deviate from the market return, and as a result, the dispersion level or variance between individuals' return and market return will be zero". Return dispersion (RD) continues to play a vital role in explaining the cross-sectional variation in expected returns, even if idiosyncratic volatility, market volatility, momentum factor, size, and book-to-market factors are included in different asset pricing models like CAPM, MVM, IVM, and FF-3 (Jiang 2010). Demirer and Jategaonkar (2013), in their study, observed a systematic conditional relation between dispersion and return even after controlling for market, size, and book-to-market factors by adding to the three Fama–French factors the return dispersion (RD) risk factor. Verousis and Voukelatos (2018) suggested that the cross-sectional dispersion (CSD) of stock returns is negatively associated with the investment, meaning that assets with high dispersion offer lower returns. Caoa et al. (2019) analyzed the relationship between the cross-sectional dispersion (CSD)

of returns and active fund performance in Australia, and concluded that outperformance occurs only for funds with high return dispersion. Last year, Qontigo presented a new index (EURO STOXX 50 realized dispersion index) to replicate a dispersion indicator on selected Eurozone blue chips "https://qontigo.com/index/sx5edisp/ (accessed on 7 July 2023)". By using a modified version of the VIX methodology, the CBOE S&P500 Dispersion Index (DSPX) was used to measure the expected dispersion in the S&P500 over the next 30 calendar days "https://www.cboe.com/us/indices/dispersion/ (accessed on 7 July 2023)". Chiang and Zheng (2010) suggested an approach to examining the dispersion measure: the cross-sectional absolute deviation of returns (CSAD). The main idea of this approach is that if the investors adhere to the same market expectations, the relationship between the cross-sectional deviation and market portfolio will be linear, and this means that the CSAD will decrease or increase, at least less than the relative rate of market return. The CSAD methodology is the primary indicator used for examining herding behavior in the next part of our study. This gives us ground to determine whether ESG leaders' portfolios tilt towards the dispersion of returns by using the CSAD approach. Based on a related literature review, the current study explores the second hypothesis:

**H2.** *ESG leader's portfolios tilt towards the volatility and dispersion of returns.*

*2.3. ESG and Herding*

An exciting and little-examined area of behavioral finance relates to herding behavior, and exploring whether herding bias can be more intense when volatility prevails in the ESG leader's market. An increased demand for ESG investing (Benz et al. 2020) may guide investors and portfolio managers towards herding bias (Przychodzen et al. 2016; Rubbaniy et al. 2021). Investors and fund managers that herd imitate other investing actions without consideration of their own fundamental analysis or private information. The herding phenomenon has been empirically investigated in various markets, such as stocks, bonds, ETFs, REITs, ESG, cryptocurrency, and, lately, artificial intelligence.

A herding methodology based on the cross-dispersion of assets performed by Hwang and Salmon (2004) captured significant herding movements in the South Korean and US markets. Using data from the Pacific Basin stock market region, Chiang et al. (2013) pointed out a lower herding bias in positive volatility index (VIX) returns. In the same way, Economou et al. (2015) indicated herding behaviors on days with negative VIX index returns in the Euronext markets. Furthermore, by employing time-varying models, Arjoon and Bhatnagar (2017) captured herding effects across two frontier markets, the Trinidad and Tobago Stock Exchange, from 2001 to 2014. As Benz et al. (2020) claimed, investors, hedge funds, and fund managers display sustainable investment herding due to concerns about adhering to the market consensus, applying data (2007–2020) from the MSCI U.S.A. ESG leader index. Rubbaniy et al. (2021) detected herding behavior during bull and bear market periods. Furthermore, Fu and Wu (2021) used the Markov regime-switching model to capture herding bias in the Chinese stock market. Regarding sustainable investing, Blondel (2022) deduced that low-risk investors herd less than medium-risk investors by surveying 175 investors for two months in 2021, whereas in relation to asset picking, passive investors herd more than active investors. Gavrilakis and Floros (2023a) recently reported herding behavior in ESG investing in Portugal, Italy, and Greece during the COVID-19 outbreak. Finally, Ameye et al. (2023) found evidence of significant herding effects in 307 firms using artificial intelligence technology, and secondly, they concluded that uncertainty moderates herding.

Contrariness, an anti-herding behavior, was noticed in data concerning metal commodities futures by Babalos and Stavroyiannis (2015). The same result was reported by Rompotis (2018), who investigated a sample of 34 small-cap and 66 large-cap ETFs from 2012 to 2016. Moreover, Coskun et al. (2020) captured anti-herding behavior in 14 leading cryptocurrencies using GARCH, Time-Varying Markov-Switching (TV-MS), and CSAD methodologies. Using the same approach, Yarovaya et al. (2021) noticed no indications of

herding bias in the cryptocurrency market. Moreover, using a database of 10.456 Global ESG funds, Ciciretti et al. (2021) found anti-herding behavior in their sample from 2012 to 2018. During the COVID-19 pandemic, anti-herding predominated (Yang and Chuang 2023) due to market uncertainty. This literature has led us to develop and examine the third hypothesis:

**H3.** *ESG leaders' portfolios lean toward herding.*

## 3. Methodology and Data

### 3.1. Regression Analysis

In this paper, we expand the literature on ESG portfolio management by analyzing the abnormal returns of selected ESG leaders' portfolios and exploring factor loadings based on the well-known Capital Asset Pricing Model (CAPM), and the Fama–French three-factor (FF3) and five-factor (FF5) models. Furthermore, we extend this analysis by introducing two additional factors into the FF5 regression model: the volatility index ($Vix_t$) and the cross-sectional absolute deviation of returns ($CSAD_t$) indicator. The above methodologies enable us to develop greater awareness concerning excess ESG returns or the alphas, risk, and return dispersion.

The first model used in the regression analysis is the Capital Asset Pricing Model (CAPM), developed by Treynor (1961):

$$ESG_{it} - R_{ft} = a_{it} + \beta_1(MKT_t) + \varepsilon_{it} \tag{1}$$

where $ESG_{it} - R_{ft}$ is the excess return of European ESG leaders portfolios; $R_{ft}$ is the one-year Treasury Bill rate; $a_{it}$ is the Jensen's alpha; $MKT_t$ is the market risk premium $\left(R_{mt} - R_{ft}\right)$ on day $t$; $\beta_1$ is the beta or the sensitivity of the $ESG_{it}$ to the market, and $\varepsilon_{it}$ is the error term. The CAPM is a suitable methodology for use in investigating the ESG performance with respect to the passive investment strategy.

The next model employed is the Fama and French (1993) three-factor model (FF3):

$$ESG_{it} - R_{ft} = a_{it} + \beta_1(MKT_t) + \beta_2(SMB_t) + \beta_3(HLM_t) + \varepsilon_{it} \tag{2}$$

where ($SMB_t$) and ($HLM_t$) are, respectively, the firm's size and value characteristics. $SMB_t$ (small minus big) represents the size premium, meaning large-cap assets are expected to earn lower returns than small-cap assets (Zehir and Aybars 2020). $HLM_t$ (high minus low) stands for the value premium; stocks with low book-to-market ratios are expected to underperform compared to those with high book-to-market ratios. The regression coefficients $\beta_1$, $\beta_2$, and $\beta_3$ explain the ESG leaders' portfolios' sensitivities to the pre-specified indicators.

Equation (3) is the Fama and French (2015) five-factor (FF5) model:

$$ESG_{it} - R_{ft} = a_{it} + \beta_1(MKT_t) + \beta_2(SMB_t) + \beta_3(HLM_t) + \beta_4(RMW_t) + \beta_5(CMA_t) + \varepsilon_{it} \tag{3}$$

where ($RMW_t$) and ($CMA_t$) are, respectively, returns for profitability and investment factors. $RMW_t$ (robust minus weak) relates to the profitability premium, meaning stocks with weak operating profitability are foreseen to underperform stocks with robust operating profitability. $CMA_t$ (conservative minus aggressive) is the return difference between stocks that invest conservatively minus those that invest aggressively. The regression coefficients $\beta_4$ and $\beta_5$ are the ESG leaders' portfolios' sensitivities to profitability and investment factors.

The effects of volatility and herding are crucial disclosures when determining abnormal portfolio returns. We use the respective implied volatility index ($Vix_t$) and the cross-sectional absolute deviation of returns ($CSAD_t$) factor to estimate the impact of the volatility and dispersion of returns on ESG leaders' portfolio performances. Therefore, we extend the FF5 model to the FF7 model, augmented by addition of the $Vix_t$ indicator and the herding $CSAD_t$ indicator, as follows:

$$ESG_{it} - R_{ft} = a_{it} + \beta_1(MKT_t) + \beta_2(SMB_t) + \beta_3(HLM_t) + \beta_4(RMW_t) + \beta_5(CMA_t) + \beta_6(Vix_t) + \beta_7(CSAD_t) + \varepsilon_{it} \quad (4)$$

We applied the traditional OLS with Newey and West's (1987) standard errors to evaluate the coefficients.

The value of $MKT_t$ is the market risk premium $\left(R_{mt} - R_{ft}\right)$, or the excess return on the market, and is calculated as the value-weighted return on all NYSE, AMEX, and NASDAQ stocks minus the one-month Treasury bill rate $(R_{ft})$. The values $MKT_t$ $SMB_t$, $HLM_t$, $RMW_t$, and $CMA_t$ in the regression models were derived from the Kenneth R. French Library (2023).

### 3.2. Herding Analysis

In checking the robustness for herding, we employed the cross-sectional absolute deviation of returns (CSAD), introduced by Chang et al. (2000), which is the most accepted method of return dispersion in the behavioral finance literature:

$$CSAD_t = \frac{1}{N}\sum_{i=1}^{N}|R_{i,t} - R_{m,t}| \quad (5)$$

where $CSAD_t$ is the return dispersion of assets on day $t$, $R_{i,t}$ is the return of the $i$th asset on day $t$, $R_{m,t}$ represents the market return on day $t$, and $N$ is the number of assets in the ESG index employed. To capture any herding behavior, we used the following model set out by Chang et al. (2000):

$$CSAD_t = \alpha + \gamma_1|R_{m,t}| + \gamma_2 R_{m,t}^2 + \varepsilon_t \quad (6)$$

where $R^2m,t$ is used for checking the non-linearity of the relationship, and $\gamma_1$ will be positive and $\gamma_2$ will be equal to zero in the absence of any herding. A significant and negative value of the parameter $\gamma_2$ suggests a herding bias since it verifies "that during a market disturbance, a nonlinear negative relationship exists between return dispersion and $R^2m,t$" (Rompotis 2018). Concerning herding asymmetries, we follow the approaches of Chang et al. (2000), Chiang and Zheng (2010), Economou et al. (2011), and Philippas et al. (2013). To evaluate the asymmetric behavior of market return dispersion, we apply Equation (7):

$$CSAD_t = a + \gamma_1 D^{up}|R_{m,t}| + \gamma_2(1 - D^{up})|R_{m,t}| + \gamma_3 D^{up}(R_{m,t})^2 + \gamma_4\left(1 - D^{up}\right)\left(R_{m,t}\right)^2 + \varepsilon_t \quad (7)$$

$D^{up}$ is a dummy variable taking a value of 1 when there are positive market returns and 0 on days with negative market returns. We run Equation (7) to estimate any herding behavior during the development of a market. In the absence of herding, both parameters $\gamma_1$ and $\gamma_2 > 0$. If $\gamma_3$ and $\gamma_4$ are negative and $\gamma_3 > \gamma_4$, then herding is more evident during negative market returns. The above models allow us to capture any herding behavior or bias in the ESG leaders' portfolios.

## 4. Results and Discussion

This section presents the results of European ESG leaders' portfolios and the outcomes of robustness tests of Global ESG leaders' index-based portfolios. Tables 1 and 2 describe the indices of selected European and Global ESG leaders. Figure 1 illustrates that the cumulative return of the benchmark index (STOXX Europe 600) significantly exceeded those of the European ESG leaders' pillars, ending in a full-sample cumulative return of 76.01% from 1 June 2012 to 15 July 2022. The performance of the STOXX Europe ESG leaders' portfolio (9.92%) is the most notable. The environmental pillar reported a 22.22% return, the social pillar displayed a 16.07% return, and the governance pillar resulted in a cumulative return of 23.20%. Figure 2 depicts the market returns of the STOXX Global 1800 index and the Global ESG pillar leaders' indices. The cumulative return of the global

benchmark index was 170.3% from 1 June 2012 to 15 July 2022. The cumulative return of STOXX Global ESG leaders was 105.42%, close to the social pillar, with 101.5% return, and the governance pillar, with 98.35% return. The environmental pillar showed a 119.41% cumulative return. The above findings show considerably higher returns on Global ESG leaders' investments than European ESG leaders' investments.

**Table 1.** Descriptions of the selected European ESG leaders' indices (Qontigo 2023).

| ESG and the Benchmark Index | Index Purpose/Descriptions from Factsheet (Qontigo 2023) |
|---|---|
| Stoxx Europe ESG Environmental Leaders Index | The STOXX Europe ESG Environmental Leaders Index aims to measure the performance of portfolios of equity that have been selected with reference to their business interests and policies toward environmental issues, and which are designed to provide exposure to certain variables known as factors; these factors can assist in the explanation of market performance. A detailed ESG report describing how these factors are applied to this index is available on the STOXX website, "www.stoxx.com/resources (accessed on 7 July 2023)". |
| Stoxx Europe ESG Social Leaders Index | The STOXX Europe ESG Social Leaders Index aims to measure the performance of portfolios of securities that have been selected with reference to their business interests and policies toward social issues, and which are designed to provide exposure to certain variables known as factors; these factors can assist in the explanation of market performance. A detailed ESG report describing how these factors are applied to this index is available on the STOXX website, "www.stoxx.com/resources (accessed on 7 July 2023)". |
| Stoxx Europe ESG Governance Leaders Index | The STOXX Europe ESG Governance Leaders Index aims to measure the performance of portfolios of equity that have been selected with reference to their business interests and policies toward governance issues, and which are designed to provide exposure to certain variables known as factors; these factors can assist in the explanation of market performance. A detailed ESG report describing how these factors are applied to this index is available on the STOXX website, "www.stoxx.com/resources (accessed on 7 July 2023)". |
| Stoxx Europe ESG Leaders Select 30 Index | The STOXX ESG Leaders Select 30 Index aims to measure the performance of portfolios of securities that have been selected with reference to their business interests and policies toward environmental, social, and governance issues. This index includes benchmarks in which the portfolios are weighted by free-float market capitalization; portfolios may also be weighted to reduce risk using volatility or variance weighting schemes. "www.stoxx.com/resources (accessed on 7 July 2023)". |
| Stoxx Europe 600 Index (Benchmark index) | The STOXX Europe 600 index is a stock index of European stocks designed by STOXX Ltd. It has 600 components representing large, mid, and small capitalization companies from 17 European countries, covering approximately 90% of the free-float market cap. "https://qontigo.com/index/sxxp/ (accessed on 7 July 2023)" |

Source: Qontigo (2023). Available at: "https://qontigo.com/ (accessed on 7 July 2023)".



**Table 2.** Descriptions of the selected Global ESG Leaders indices (Qontigo 2023).

| ESG and the Benchmark Index | Index Purpose/Descriptions from Factsheet (Qontigo 2023) |
|---|---|
| Stoxx Global ESG Environmental Leaders Index | The STOXX Global ESG Environmental Leaders index provides access to global environmental leaders through quantitative selection. The sustainability data in environmental areas are supplied by Sustainalytics ("https://www.sustainalytics.com/esg-data (accessed on 7 July 2023)". The index follows a bottom-up approach based on the company's ESG scores. The system ranges from 0 to 100 points and is applied to the environmental pillar. Index components are weighted according to their ESG scores "www.stoxx.com/indices/rulebooks.html (accessed on 7 July 2023)". |
| Stoxx Global ESG Social Leaders Index | The STOXX Global ESG Social Leaders index provides access to global social leaders through quantitative selection. The sustainability data in social areas are supplied by Sustainalytics "https://www.sustainalytics.com/esg-data (accessed on 7 July 2023)". The index follows a bottom-up approach based on the company's ESG scores. The system ranges from 0 to 100 points and is applied to the social pillar. Index components are weighted according to their ESG scores "www.stoxx.com/indices/rulebooks.html (accessed on 7 July 2023)". |
| Stoxx Global ESG Governance Leaders Index | The STOXX Global ESG Governance Leaders index provides access to global governance leaders through quantitative selection. The sustainability data in governance areas are supplied by Sustainalytics (https://www.sustainalytics.com/esg-data). The index follows a bottom-up approach based on the company's ESG scores. The system ranges from 0 to 100 points and is applied to the governance pillar. Index components are weighted according to their ESG scores "www.stoxx.com/indices/rulebooks.html (accessed on 7 July 2023)". |
| Stoxx Global ESG Leaders Index | The STOXX Global ESG Leaders' indices consist of one broad and three specialized indexes for environmental, social, and governance pillars. The three specialized indices form the broad STOXX Global ESG Leaders Index. The indices provide access to global sustainability leaders through quantitative selection. The sustainability data in environmental, social, and governance areas are supplied by Sustainalytics "(https://www.sustainalytics.com/esg-data (accessed on 7 July 2023)". |
| Stoxx Global 1800 Index (Benchmark index) | The STOXX Global 1800 Index contains 600 European, 600 North American, and 600 Asia/Pacific region stocks represented by the STOXX Europe 600 Index, the STOXX North America 600 Index, and the STOXX Asia/Pacific 600 Index. "https://qontigo.com/index/sxw1e/ (accessed on 7 July 2023)". |

Source: Qontigo (2023). Available at: "https://qontigo.com/ (accessed on 7 July 2023)".

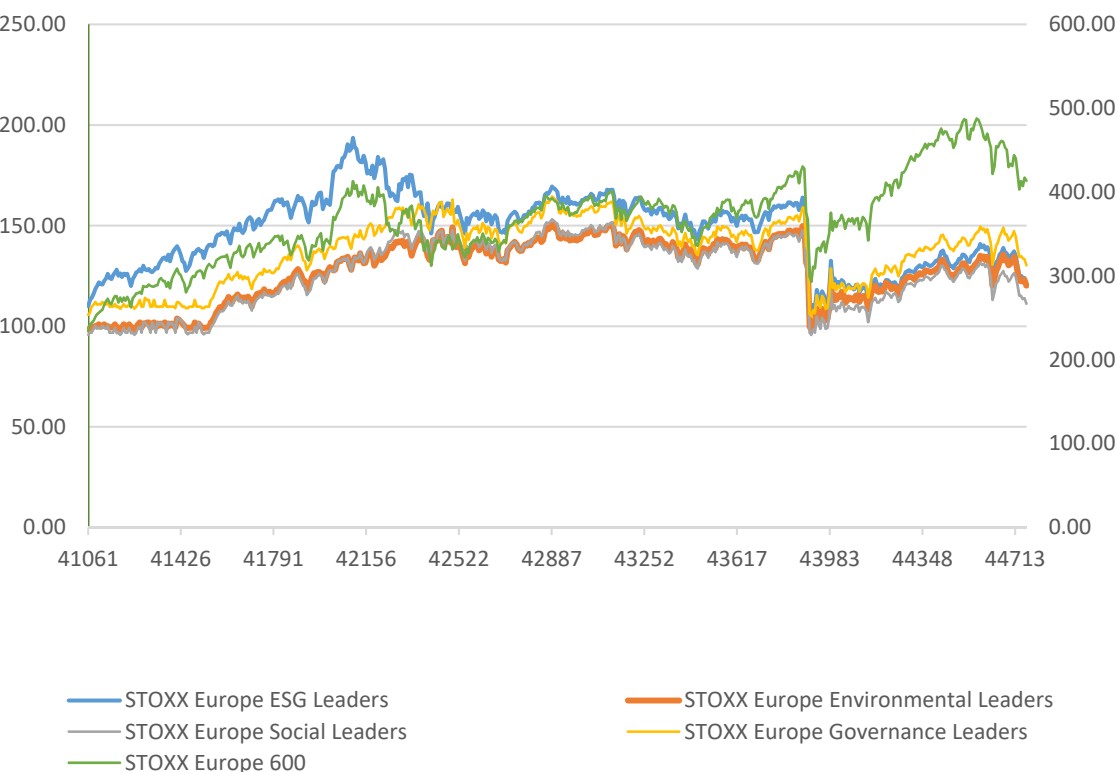

**Figure 1.** Cumulative returns of Europe ESG pillars leaders related to the benchmark (STOXX Europe 600).

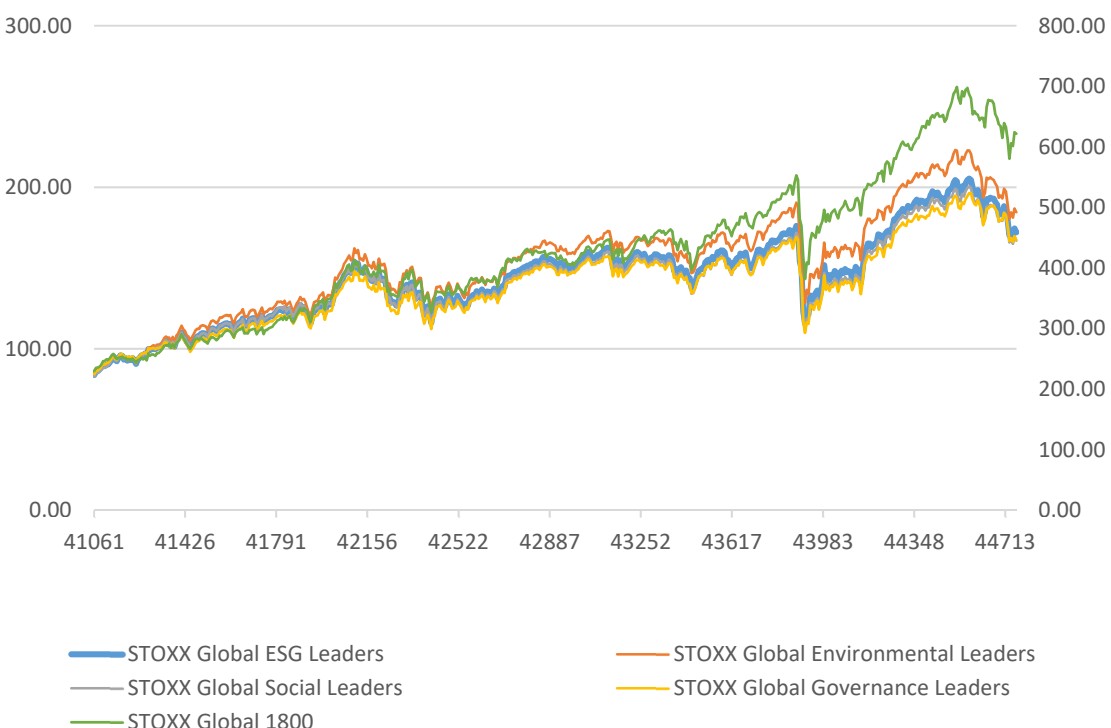

**Figure 2.** Cumulative returns of Global ESG pillars leaders related to the benchmark (STOXX Global 1800) index.

Tables 3 and 4 show the correlation matrices of selected European and Global ESG pillars, along with the volatility and dispersion of returns factor. All indices are highly positively correlated, while the series is less positively correlated with the volatility index and slightly negatively correlated with the dispersion of returns index, implying that

when the dispersion of returns grows, the ESG leaders' performances decline, which strongly adheres to the risk–return theory. Tables 5 and 6 depict the descriptive statistics of European and Global ESG leaders' portfolios. The results illustrate, on average, a significant dispersion across all ESG leaders' indices. The annual mean standard deviation (volatility) suggests that ESG leaders' investing indicates low risk and negative asymmetry. The summary statistics show the daily returns' negative asymmetry (negative skewness values). The distributions of ESG indices are leptokurtic, as the kurtosis values are above 12, indicating the presence of fat tails in the series. Kurtosis is only valid when used in connection with standard deviation. ESG indices have high kurtosis, which is unsuitable for investment strategies as the returns are close to the mean, but the overall volatility of the ESG indices is low, which should be considered. The distributions are non-normal for all series as the Jarque–Bera statistics show significant *p*-values, indicating the departure from normality and the arrival of volatility clustering. In some cases, we apply logarithms into our series to overcome non-normality. The Augmented Dickey–Fuller (ADF) test results have significant *p*-values. As a result, we reject the H0 hypothesis (the series have a unit root problem) and accept the stationarity of the series under study. This finding aligns with that of Gavrilakis and Floros (2023b), who concluded that the ESG/thematic market does not have a random walk (non-normal) distribution and that predicting a long-term relationship between the series' elements is feasible. In the same study, the authors employed several Copula models to reproduce asymmetries in the asymptotic tail dependence of stationary ESG/thematic investment. Copulas have been used extensively in quantitative finance to capture tail risk and, lately, in applications for portfolio optimization (Patton 2012; Dewick and Liu 2022; Nagler et al. 2022). As suggested by many studies on ESG investing that support the null hypothesis of stationarity (Jain et al. 2020; Górka and Kuziak 2021; Ouchen 2022; Erol et al. 2023), we now proceed by applying an econometric analysis to the reference series.

**Table 3.** Correlation matrix of selected European ESG Leaders indices.

| | Stoxx Europe ESG Environmental Leaders Index | Stoxx Europe ESG Social Leaders Index | Stoxx Europe ESG Governance Leaders Index | Stoxx Europe ESG Leaders Select 30 Index | Stoxx Europe 600 Index | *VIXt* | *CSADt* |
|---|---|---|---|---|---|---|---|
| Stoxx Europe ESG Environmental Leaders Index | 1 | | | | | | |
| Stoxx Europe ESG Social Leaders Index | 0.9826 | 1 | | | | | |
| Stoxx Europe ESG Governance Leaders Index | 0.9868 | 0.9839 | 1 | | | | |
| Stoxx Europe ESG Leaders Select 30 Index | 0.7528 | 0.7577 | 0.7532 | 1 | | | |
| Stoxx Europe 600 Index | 0.7474 | 0.7433 | 0.7410 | 0.7507 | 1 | | |
| *VIXt* | 0.1389 | 0.1377 | 0.1392 | 0.1504 | 0.1846 | 1 | |
| *CSADt* | −0.0659 | −0.0665 | −0.0635 | −0.0735 | −0.0330 | −0.0203 | 1 |

**Table 4.** Correlation matrix of selected Global ESG Leaders indices.

| | Stoxx Global ESG Environmental Leaders Index | Stoxx Global ESG Social Leaders Index | Stoxx Global ESG Governance Leaders Index | Stoxx Global ESG Leaders Index | Stoxx Global 1800 Index | VIXt | CSADt |
|---|---|---|---|---|---|---|---|
| Stoxx Global ESG Environmental Leaders Index | 1 | | | | | | |
| Stoxx Global ESG Social Leaders Index | 0.9938 | 1 | | | | | |
| Stoxx Global ESG Governance Leaders Index | 0.9908 | 0.9950 | 1 | | | | |
| Stoxx Global ESG Leaders Select Index | 0.9967 | 0.9981 | 0.9972 | 1 | | | |
| Stoxx Global 1800 Index | 0.6572 | 0.6453 | 0.6310 | 0.6463 | 1 | | |
| VIXt | 0.1906 | 0.1860 | 0.1892 | 0.1900 | 0.0746 | 1 | |
| CSADt | −0.1338 | −0.1331 | −0.1357 | −0.1347 | −0.0913 | −0.0611 | 1 |

**Table 5.** Descriptive statistics of European ESG Leaders indices.

| Statistic | Stoxx Europe ESG Environmental Leaders Index | Stoxx Europe ESG Social Leaders Index | Stoxx Europe ESG Governance Leaders Index | Stoxx Europe ESG Leaders Select 30 Index | Stoxx Europe 600 Index |
|---|---|---|---|---|---|
| Min | −0.1381 | −0.1419 | −0.1383 | −0.1460 | 0.1147 |
| Max | 0.0778 | 0.0766 | 0.0793 | 0.0860 | 0.0840 |
| Mean | $6.8668 \times 10^5$ | $4.4887 \times 10^5$ | $6.1835 \times 10^5$ | $-2.7927 \times 10^5$ | 0.0002 |
| SD | 0.0089 | 0.0091 | 0.0091 | 0.0106 | 0.0103 |
| Skewness | −2.2557 | −2.1594 | −2.2102 | −1.3565 | −0.8440 |
| Kurtosis | 34.012 | 33.498 | 32.635 | 20.774 | 11.112 |
| JB test | 124,832 *** | 120,966 *** | 115,009 *** | 46,564 *** | 13,400.6 *** |
| ADF test | −13.4561 *** | −12.4959 *** | −13.236 *** | −18.4363 *** | −18.1139 *** |

Normality test based on skewness, kurtosis values, Jarque–Bera test, and Augmented Dickey–Fuller test, *** $p < 0.01$.

**Table 6.** Descriptive statistics of Global ESG Leaders indices.

| Statistic | Stoxx Global ESG Environmental Leaders Index | Stoxx Global ESG Social Leaders Index | Stoxx Global ESG Governance Leaders Index | Stoxx Global ESG Leaders Index | Stoxx Global 1800 Index |
|---|---|---|---|---|---|
| Min | −0.1069 | −0.1125 | −0.1131 | −0.1108 | −0.0983 |
| Max | 0.0881 | 0.0959 | 0.0916 | 0.0919 | 0.0826 |
| Mean | 0.0002 | 0.0001 | 0.0001 | 0.0002 | 0.0004 |
| SD | 0.0098 | 0.0102 | 0.0101 | 0.0100 | 0.0094 |
| Skewness | −0.7180 | −0.7054 | −0.7848 | −0.7432 | −0.8955 |
| Kurtosis | 12.384 | 13.265 | 13.270 | 13.066 | 14.453 |
| JB test | 16,488.4 *** | 18,878.4 *** | 18,935.3 *** | 18,345.6 *** | 22,489.9 *** |
| ADF test | −16.1624 *** | −16.0008 *** | −16.0273 *** | −16.0829 *** | −13.1973 *** |

Normality test based on skewness, kurtosis values, Jarque–Bera test, and Augmented Dickey–Fuller test, *** $p < 0.01$.

Table 7 illustrates the OLS regression results (with HAC standard errors) of European ESG Leaders' index-based portfolios. The validity of the OLS model was tested based on multicollinearity tests detecting values below ten. Therefore, there is no evidence of a multicollinearity problem. The results of all factor models suggest that all the ESG leaders' portfolios underperformed (with negative risk-adjusted abnormal returns) against the market. We confirm the findings of Mollet and Ziegler's (2014) study, which identified insignificant abnormal returns on three different portfolios ("MSCI sustainability leaders", "Sustainability leaders", and "Other MSCI firms") for Europe and the US from 1998 to 2009. Irrespective of our sample, the ESG portfolios present medium betas ($\beta$), indicating a medium level of systematic risk. The ESG portfolios depict a negative loading on $SMB_t$, implying a tilt toward large-cap firms. The factor loadings for the determinant $HLM_t$ are broadly significantly positive in all series, which indicates a bias towards value stocks. The ESG portfolios report a favorable loading on $RMW_t$, indicating a tilt toward robust operating profitability. The statistically significant and positive $CMA_t$ exposure suggests that the ESG portfolios include firms with conservative investment strategies, usually associated with low future returns. The factor loadings are homogeneous across the environmental, social, and governance components. The ESG premium remains statistically significant after adjusting for the CAPM, FF3, FF5, and augmented FF7 models. This finding supports the results in the literature that FF3 and FF5 effectively explain ESG leaders' performance concerning market returns (Zaremba and Czapkiewicz 2017; Guo et al. 2017). Furthermore, the results confirm that our augmented FF7 model significantly explains all the factor loadings. Table 7 also reports that the fear or volatility index ($Vix_t$) is significant and positive in all ESG indices, indicating a volatility bias. We verify the findings of the study of Górka and Kuziak (2021), who confirmed a higher dependence on the volatility of selected ESG indices from 2007 until 2019. Finally, the $CSAD_t$ factor is significant and negative in European ESG Leaders' portfolios, suggesting a dispersion of return bias. To check the robustness of our results, we rerun the FF7 OLS model by adding the VSTOXX index instead of VIX, ending up with the same results (see Table 8) but with minor differences in the importance of loading factors ($CSAD_t$).

**Table 7.** Regression equation table for European ESG Leaders index-based portfolios. The regression results of STOXX Europe index-based ESG pillars against CAPM, FF3, FF5, and FF7 (including $VIX_t$ and $CSAD_t$ in the FF5 model). Daily price data were obtained from the Refinitive Eikon database (2022) from 1 June 2012 to 14 July 2022. The model FF7 takes the form: $ESG_{it} - R_{ft} = a_{it} + \beta_1(MKT_t) + \beta_2(SMB_t) + \beta_3(HLM_t) + \beta_4(RMW_t) + \beta_5(CMA_t) + \beta_5(Vix_t) + \beta_5(CSAD_t) + \varepsilon_{it}$.

| Model | | Stoxx Europe ESG Environmental Leaders | Stoxx Europe ESG Social Leaders | Stoxx Europe ESG Governance Leaders | Stoxx Europe ESG Leaders Select 30 | Stoxx Europe 600 |
|---|---|---|---|---|---|---|
| CAPM | $\alpha$ | −0.0021 *** | −0.0022 *** | −0.0021 *** | −0.0023 *** | −0.0020 *** |
| | $\beta$ | 0.00601 *** | 0.0062 *** | 0.0060 *** | 0.0092 *** | 0.0083 *** |
| | $R^2$ | 0.4376 | 0.4447 | 0.43141 | 0.748 | 0.6467 |
| FF3 | $\alpha$ | −0.0021 *** | −0.0021 *** | −0.0021 *** | −0.0021 *** | −0.0020 *** |
| | $\beta$ | 0.0053 *** | 0.0056 *** | 0.0054 *** | 0.0059 *** | 0.0074 *** |
| | $SMBt$ | −0.0021 *** | −0.0017 *** | −0.0020 *** | −0.0046 *** | −0.0044 *** |
| | $HMLt$ | 0.00151 *** | 0.0018 *** | 0.0017 *** | 0.0009 *** | −0.0004 * |
| | $R^2$ | 0.453 | 0.460 | 0.4486 | 0.5360 | 0.6687 |

**Table 7.** *Cont.*

| Model | | Stoxx Europe ESG Environmental Leaders | Stoxx Europe ESG Social Leaders | Stoxx Europe ESG Governance Leaders | Stoxx Europe ESG Leaders Select 30 | Stoxx Europe 600 |
|---|---|---|---|---|---|---|
| FF5 | $\alpha$ | −0.0022 *** | −0.0022 *** | −0.0022 *** | −0.0022 *** | −0.0020 *** |
| | $\beta$ | 0.0056 *** | 0.0058 *** | 0.0056 *** | 0.0062 *** | 0.0075 *** |
| | $SMBt$ | −0.0018 *** | −0.0014 *** | −0.0016 *** | −0.0041 *** | −0.0043 *** |
| | $HMLt$ | 0.0023 *** | 0.0028 *** | 0.0027 *** | 0.0016 *** | −0.0003 |
| | $RMWt$ | 0.0042 *** | 0.0049 *** | 0.0049 *** | 0.0050 *** | 0.0005 |
| | $CMAt$ | 0.0029 *** | 0.0029 *** | 0.0031 *** | 0.0043 *** | 0.0009 |
| | $R^2$ | 0.4635 | 0.4731 | 0.4615 | 0.5501 | 0.6694 |
| FF7 | $\alpha$ | −0.0018 | −0.0019 *** | −0.0019 *** | −0.0016 *** | −0.0023 *** |
| | $\beta$ | 0.0054 *** | 0.0057 *** | 0.0055 *** | 0.0060 *** | 0.0073 *** |
| | $SMBt$ | −0.0020 *** | −0.0016 *** | −0.0019 *** | −0.0046 *** | −0.0046 *** |
| | $HMLt$ | 0.0024 *** | 0.0030 *** | 0.0029 *** | 0.0019 *** | $-4.16848 \times 10^6$ |
| | $RMWt$ | 0.0044 *** | 0.0050 *** | 0.0050 *** | 0.0052 *** | 0.0008 |
| | $CMAt$ | 0.0027 *** | 0.0028 *** | 0.0029 *** | 0.0040 *** | 0.0006 |
| | $VIXt$ | 0.0053 *** | 0.0052 *** | 0.0057 *** | 0.0095 *** | 0.0098 *** |
| | $CSADt$ | −0.0376 * | −0.0371 * | −0.0333 | −0.0686 ** | 0.0300 |
| | $R^2$ | 0.4653 | 0.4746 | 0.4634 | 0.5555 | 0.6742 |

Note: ***, ** and * denote statistical significance at the 1%, 5%, and 10% levels, respectively. The risk-free rate $(R_{ft})$ and the values $MKT_t$, $SMB_t$, $HLM_t$, $RMW_t$, and $CMA_t$ for the regression models were derived from the data in the Kenneth R. French Library. Alpha (*a*) and beta (*b*) are expressed in basis units (e.g., an alpha of −0.0021 is −0.21%, while a beta of 0.00601 is a 0.60 beta score). The $R^2$ is adjusted and describes the goodness of fit of the model.

**Table 8.** Regression equation table for European ESG leaders' index-based portfolios. The regression results of STOXX Europe index-based ESG pillars against CAPM, FF3, FF5, and FF7 (including $VSTOXX_t$ and $CSAD_t$ in the FF5 model). Daily price data were obtained from the Refiitive Eikon database (2022) from 1 June 2012 to 14 July 2022. The model FF7 takes the form: $ESG_{it} - R_{ft} = a_{it} + \beta_1(MKT_t) + \beta_2(SMB_t) + \beta_3(HLM_t) + \beta_4(RMW_t) + \beta_5(CMA_t) + \beta_5(VSTOXX_t) + \beta_5(CSAD_t) + \varepsilon_{it}$.

| Model | | Stoxx Europe ESG Environmental Leaders | Stoxx Europe ESG Social Leaders | Stoxx Europe ESG Governance Leaders | Stoxx Europe ESG Leaders Select 30 | Stoxx Europe 600 |
|---|---|---|---|---|---|---|
| FF7 | $\alpha$ | −0.0017 *** | −0.0018 *** | −0.0018 *** | −0.0021 *** | −0.0022 *** |
| | $\beta$ | 0.0042 *** | 0.0045 *** | 0.0044 *** | 0.0093 *** | 0.0084 *** |
| | $SMBt$ | −0.0013 *** | −0.0010 *** | −0.0012 *** | −0.0004 * | −0.0035 *** |
| | $HMLt$ | 0.0027 *** | 0.0032 *** | 0.0031 *** | 0.0016 *** | 0.00038 |
| | $RMWt$ | 0.0048 *** | 0.0054 *** | 0.0054 *** | 0.0033 *** | 0.0013 ** |
| | $CMAt$ | 0.0025 *** | 0.0026 *** | 0.0027 *** | 0.0037 *** | 0.0003 |
| | $VSTOXXt$ | 0.0011 *** | 0.0010 *** | −0.0010 *** | 0.0001 *** | −0.0017 *** |
| | $CSADt$ | −0.0423 | −0.0416 * | −0.0376 * | 0.0309 * | 0.0426 |
| | $R^2$ | 0.4837 | 0.4902 | 0.4780 | 0.7640 | 0.7086 |

Note: ***, **, * denote statistical significance at the 1%, 5%, and 10% levels, respectively. The risk-free rate $(R_{ft})$ and the values $MKT_t$, $SMB_t$, $HLM_t$, $RMW_t$, and $CMA_t$ for the regression models were derived from the Kenneth R. French data Library. Alpha (*a*) and beta (*b*) are expressed in basis units (e.g., an alpha of −0.0017 is −0.17%, while a beta of 0.0042 is a 0.42 beta score). The $R^2$ is adjusted and describes the goodness of fit of the model.

We employed the cross-sectional absolute deviation of returns (Chang et al. 2000) methodology to enhance our results concerning the dispersion of returns bias and capture any herding effect. Table 9 summarizes our empirical results concerning herding existence in ESG leaders' portfolios. The benchmark methodology of herding presented by Chang et al. (2000) is applied in model I. Parameter $\gamma_2$ is significant and negative, implying the herding effect. This critical finding aligns with those of Benz et al. (2020), Blondel (2022) and Gavrilakis and Floros (2023a), who confirmed herding behavior in their studies related to ESG investing. ESG leaders' investors are involved in herd behavior during flat markets, which may result in market inefficiency and less diversified portfolios. Finally, our outcomes do not indicate herding behavior for up- or down-market days, as coefficients $\gamma_1$ and $\gamma_2 > 0$. This is contrary to the findings of Rubbaniy et al. (2021), who captured herding behavior during bull and bear market periods. This anti-herding behavior probably indicates that highly ESG-scoring assets conduce to market efficiency by lowering the probability of forming a financial bubble. Figure 3 presents the effects of herding on STOXX Europe ESG leaders' portfolios. There is a negative correlation between the dispersion of returns and the performance of ESG leaders' indices, which means that a 1-unit change in ESG leaders' returns leads to a $-0.79$-unit change in the dispersion of returns (see Figure 3 and Table 9). We agree with Verousis and Voukelatos (2018), who found a negative relation between the dispersion of returns and investment performance.

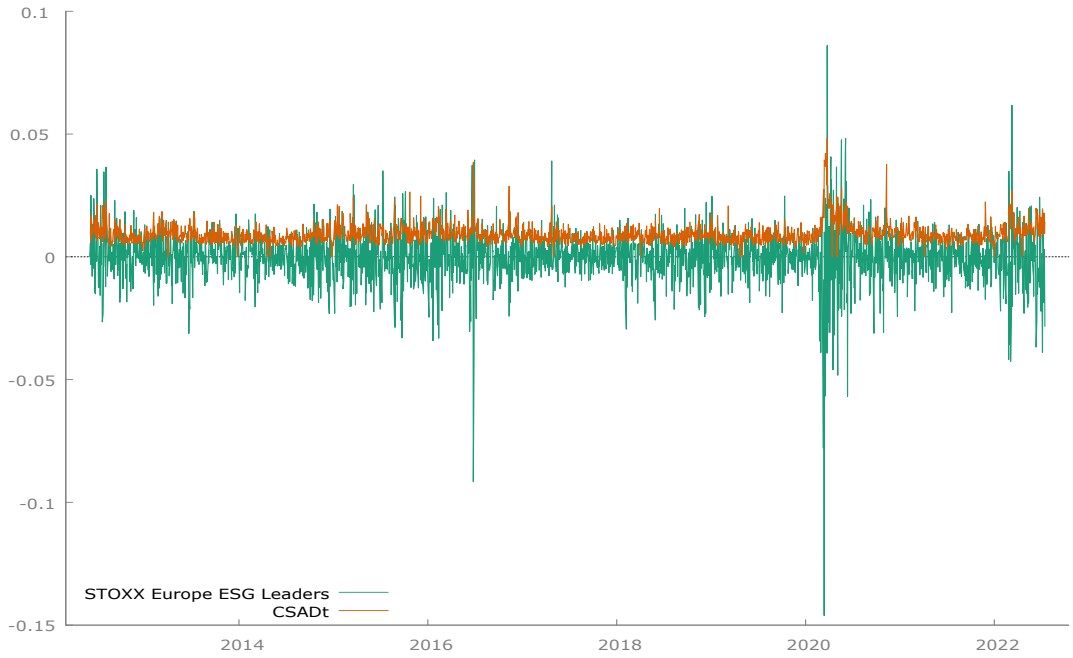

**Figure 3.** Herding in STOXX Europe ESG leaders.

To check the robustness of our empirical results for European ESG leaders, we ran the same tests but with Global ESG leaders' portfolio data. Table 10 reports the OLS (with HAC standard errors) robustness regression results for Global ESG leaders' index-based portfolios. The validity of the OLS model was tested based on multicollinearity tests detecting values below ten. Therefore, there is no evidence of a multicollinearity problem. The findings of all the factor models used suggest that the Global ESG leaders' portfolios underperformed in the market from 1 June 2012 to 15 July 2022. This argument confirms the findings of Hartzmark and Sussman (2019), who found causal evidence that investors value sustainability, but no evidence of positive effects on performance. In addition, we partly confirm the findings of Auer and Schuhmacher (2016), who argued that high- or low-rated ESG stock selection does not provide investors with superior returns, while the returns for Asia-Pacific and the US are similar to ESG performance, but for European markets, they are lower. Moreover, all the ESG portfolios show medium betas ($\beta$), indicating a

medium level of systematic risk. The Global ESG portfolios presented a positive loading on $SMB_t$, but this was not significant, except for in the social portfolio, implying a tilt towards small firms. The factor loadings for the determinant $HLM_t$ are significantly positive in all series, which indicates a bias towards value stocks. The $RMW_t$ and $CMA_t$ exposure are not statistically significant in all factor models. In summary, the factor loadings are similar across the environmental, social, and governance components. According to our results, the volatility index ($Vix_t$) is significant and positive in all portfolios, indicating a volatility bias. This finding agrees with those of Nishant et al. (2022), who argued that global companies with better ESG scores are more resilient in high-volatility environments. Furthermore, the $CSAD_t$ indicator is significant and negative in all portfolios, suggesting a dispersion of returns bias.

Once more, we employed the cross-sectional absolute deviation of returns to attest to the dispersion of returns bias and capture any herding effects (Chang et al. 2000). Model I in Table 11 shows evidence of herding, since coefficient $\gamma_2$ is statistically significant and negative, while our empirical results do not indicate herding behavior toward the ESG leaders' returns on up- or down-market days, as coefficients $\gamma_1$ and $\gamma_2 > 0$. Overall, our findings align with those of Rubbaniy et al. (2021) and Fu and Wu (2021), who confirmed herding behavior in their studies related to Global ESG investing. We do not confirm the findings of Ciciretti et al. (2021), who captured no herding behavior in 10,456 global ESG funds. Figure 4 presents the effects of herding on the STOXX Global ESG leaders' portfolio. There is a negative correlation between the dispersion of returns and the performance of ESG leader's indices, which means that a 1-unit change in ESG leaders' returns leads to a $-0.13$-unit change in the dispersion of returns (see Figure 4 and Table 11). We contradict the conclusions of Caoa et al. (2019), who found a positive relation between fund performance and return dispersion.

**Table 9.** Results of herding models for Europe. This table indicates the results concerning the estimated coefficients of 2 fixed-effect herding models for STOXX Europe ESG Leaders Select 30 using a dataset from 1 June 2012 to 14 July 2022. Newey and West's (1987) standard errors were used. Model I: $CSAD_t = a + \gamma_1 |R_{m,t}| + \gamma_2 (R_{m,t})^2 + \varepsilon_t$; Model II: $CSAD_t = a + \gamma_1 D^{up} |R_{m,t}| + \gamma_2 (1 - D^{up}) |R_{m,t}| + \gamma_3 D^{up} (R_{m,t})^2 + \gamma_4 (1 - D^{up}) (R_{m,t})^2 + \varepsilon_t$.

| *Parameter* | *Model I* | *Model II* |
|:---:|:---:|:---:|
| *R-squared* | 0.3867 | 0.3954 |
| *Adj. R-squared* | 0.3862 | 0.3945 |
| *Log-likelihood* | 11,077.03 | 11,095.34 |
| $a$ | 0.0068 *** | 0.0070 *** |
| $\gamma_1$ | 0.3540 *** | 0.2893 *** |
| $\gamma_2$ | $-0.7990$ ** | 0.3468 *** |
| $\gamma_3$ | | 1.8071 *** |
| $\gamma_4$ | | $-0.9875$ *** |
| | Herding ($\gamma_2 < 0$) | Anti-herding ($\gamma_3 > 0$, $\gamma_4 < 0$) |

Note: ***, **, denote statistical significance at the 1% and 5% levels, respectively.

**Table 10.** Robustness regression equation table for Global ESG leaders index-based portfolios. The regression results of STOXX Global index-based ESG pillars against CAPM, FF3, FF5, and FF7 (including $VIX_t$ and $CSAD_t$ in the FF5 model). Daily price data were obtained from Refinitive Eikon database (2022) from 1 June 2012 to 14 July 2022. The model FF7 takes the form: $ESG_{it} - R_{ft} = a_{it} + \beta_1(MKT_t) + \beta_2(SMB_t) + \beta_3(HLM_t) + \beta_4(RMW_t) + \beta_5(CMA_t) + \beta_5(Vix_t) + \beta_5(CSAD_t) + \varepsilon_{it}$.

| Model | | Stoxx Global ESG Environmental Leaders | Stoxx Global ESG Social Leaders | Stoxx Global ESG Governance Leaders | Stoxx Global ESG Leaders | Stoxx Global 1800 Index |
|---|---|---|---|---|---|---|
| CAPM | $\alpha$ | −0.0022 *** | −0.0022 *** | −0.0022 *** | −0.0022 *** | −0.0021 *** |
| | $\beta$ | 0.0059 *** | 0.0061 *** | 0.0059 *** | 0.0060 *** | 0.0075 ** |
| | $R^2$ | 0.3963 | 0.3897 | 0.3663 | 0.3854 | 0.6890 |
| FF3 | $\alpha$ | −0.0022 *** | −0.0022 *** | −0.0022 *** | −0.0022 *** | −0.0021 *** |
| | $\beta$ | 0.0059 *** | 0.0061 *** | 0.0059 *** | 0.0059 *** | 0.0076 *** |
| | $SMB_t$ | 0.0003 | 0.0005 * | 0.0003 | 0.0004 | −0.0006 *** |
| | $HML_t$ | 0.0025 *** | 0.0030 *** | 0.0031 *** | 0.0029 *** | 0.0006 *** |
| | $R^2$ | 0.4359 | 0.4444 | 0.4247 | 0.4365 | 0.6934 |
| FF5 | $\alpha$ | −0.0022 *** | −0.0022 *** | −0.0022 *** | −0.002 *** | −0.0021 *** |
| | $\beta$ | 0.0059 *** | 0.0060 *** | 0.0058 *** | 0.0059 *** | 0.0076 *** |
| | $SMB_t$ | 0.0004 | 0.0005 * | 0.0003 | 0.0004 | −0.0006 *** |
| | $HML_t$ | 0.0025 *** | 0.0030 *** | 0.0031 *** | 0.0029 *** | 0.0008 |
| | $RMW_t$ | −0.0003 | −0.0005 | −0.0005 | −0.0004 | $2.41469 \times 10^5$ |
| | $CMA_t$ | −0.0002 | $1.56867 \times 10^5$ | 0.0001 | $-5.11484 \times 10^6$ | −0.0002 |
| | $R^2$ | 0.4360 | 0.4448 | 0.4250 | 0.4367 | 0.6927 |
| FF7 | $\alpha$ | −0.0016 *** | −0.0016 *** | −0.0016 *** | −0.0016 *** | −0.0021 *** |
| | $\beta$ | 0.0059 *** | 0.0061 *** | 0.0059 *** | 0.0059 *** | 0.0076 *** |
| | $SMB_t$ | 0.0003 | 0.0005 * | 0.0003 | 0.0004 | −0.0006 *** |
| | $HML_t$ | 0.0025 *** | 0.0029 *** | 0.0030 *** | 0.0028 *** | 0.0007 *** |
| | $RMW_t$ | −0.0002 | −0.0004 | −0.0004 | −0.0003 | $6.75079 \times 10^5$ |
| | $CMA_t$ | −0.0002 | $-3.72079 \times 10^5$ | 0.0001 | $-5.96169 \times 10^5$ | −0.0002 |
| | $VIX_t$ | 0.0249 *** | 0.0250 *** | 0.0252 *** | 0.0251 *** | 0.0117 *** |
| | $CSAD_t$ | −0.1040 *** | −0.1075 *** | −0.1139 *** | −0.1085 *** | −0.0063 |
| | $R^2$ | 0.4766 | 0.4836 | 0.4649 | 0.4770 | 0.7033 |

Note: ***, **, * denote statistical significance at the 1%, 5%, and 10% levels, respectively. The risk-free rate $(R_{ft})$ and the values $MKT_t$, $SMB_t$, $HLM_t$, $RMW_t$, and $CMA_t$ for the regression models were derived from the Kenneth R. French data Library. Alpha (*a*) and beta (*b*) are expressed in basis units (e.g., an alpha of −0.0022 is −0.22%, while a beta of 0.0059 is a 0.59 beta score). The $R^2$ is adjusted and describes the goodness of fit of the model.

**Table 11.** Robustness results of global herding models. This table shows the estimated coefficients of 2 fixed-effect herding models for Stoxx Global ESG leaders using a dataset from 1 June 2012 to 14 July 2022. Newey and West's (1987) standard errors were used. Model I: $CSAD_t = a + \gamma_1|R_{m,t}| + \gamma_2(R_{m,t})^2 + \varepsilon_t$. Model II: $CSAD_t = a + \gamma_1 D^{up}|R_{m,t}| + \gamma_2(1 - D^{up})|R_{m,t}| + \gamma_3 D^{up}(R_{m,t})^2 + \gamma_4(1 - D^{up})(R_{m,t})^2 + \varepsilon_t$.

| Parameter | Model I | Model II |
|---|---|---|
| *R-squared* | 0.7430 | 0.7435 |
| *Adj. R-squared* | 0.7428 | 0.7430 |
| *Log-likelihood* | 11,484.76 | 11,486.92 |
| $a$ | 0.0020 *** | 0.0020 *** |
| $\gamma_1$ | 0.6502 *** | 0.6529 *** |
| $\gamma_2$ | −0.1352 * | 0.6407 *** |
| $\gamma_3$ | | 0.2077 |
| $\gamma_4$ | | −0.1568 |
| | Herding ($\gamma_2 < 0$) | Anti-herding ($\gamma_3 > 0$, $\gamma_4 < 0$) |

Note: ***, * denote statistical significance at the 1% and 10% levels, respectively.

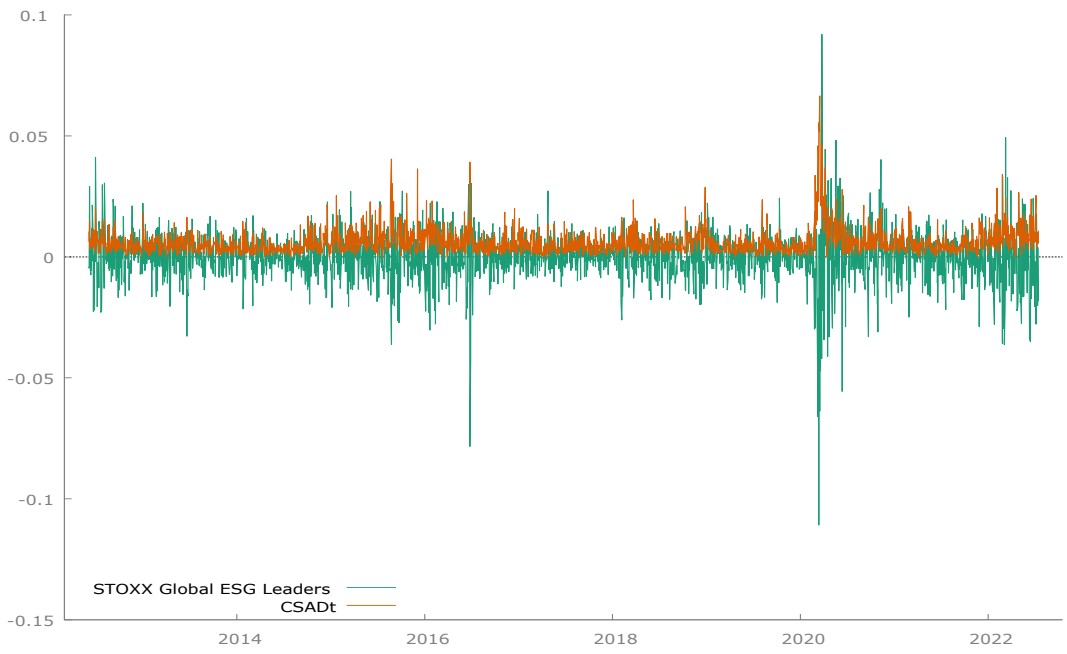

**Figure 4.** Herding in STOXX Global ESG leaders.

## 5. Conclusions

ESG investing is prevalent in today's global markets, driven by rising demand for investments that promote sustainability. Investors, fund managers, and regulators require more disclosure to evaluate the extent to which ESG leaders impact portfolio performance. We here investigated the risk-adjusted performance of ESG leaders' portfolios in European and Global ESG leaders' markets from 2012 to mid-2022. The ESG leaders' portfolios did not produce significant alphas, verifying the findings of studies on negative excess returns. The regression models CAPM, FF3, and FF5 were used to test the loading factors' validity and interpret the returns' cross-section. We noticed sufficient evidence that the European ESG leaders tilt towards large caps, and value stocks, robust operating profitability, and low-risk investment strategies. In contrast, we reported no significant evidence in relation to Global ESG leaders regarding size, operation profitability, and investment strategy. Our results provide an effective means to capture ESG leaders' abnormal returns by quantifying

the attribution of the ESG pillars' loading factors. Furthermore, we examined the effects and the validity of adding two secondary behavioral determinants (volatility and dispersion of returns) to the FF5 model, resulting in a novel finding of volatility tilting and a dispersion of returns bias on European and Global ESG leaders' portfolios. This finding has practical implications for investors and fund managers exposed to ESG leaders' assets in managing ESG funds and constructing sustainable portfolios. Finally, the indication of herding behavior in ESG leaders' investing eliminates the diversification benefits, leading to a risk exposure that would be difficult to hedge.

The current study contributes to the literature by providing valuable updates on factor loadings of different regression models on ESG leaders' portfolios. Furthermore, using well-known regression equation methodologies, it analyzes how those portfolios are affected by volatility, the cross-sectional dispersion of returns, and herding behavior, providing helpful insights that will help investors and policymakers to better understand pricing anomalies and behavioral finance. A limitation to be acknowledged is that the current study did not use herding estimation for sub-periods, time-varying betas of herding measures, or cross-market herding. Future research could examine how ESG investing can be applied for portfolio optimization. An interesting avenue would be to analyze and compare ESG leaders and laggards.

**Author Contributions:** Conceptualization, N.G. and C.F.; methodology, N.G.; software, N.G.; validation, N.G. and C.F.; formal analysis, C.F.; investigation, N.G.; resources, N.G.; data curation, N.G.; writing—original draft preparation, N.G.; writing—review and editing, C.F.; visualization, N.G.; supervision, C.F.; project administration, C.F.; All authors have read and agreed to the published version of the manuscript.

**Funding:** This research received no external funding.

**Data Availability Statement:** Data are available upon request.

**Conflicts of Interest:** The authors declare no conflict of interest.

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
