# Peer review of "Volatility and Herding Bias on ESG Leaders’ Portfolios Performance"

_jrfm, doi:10.3390/jrfm17020077_

Round 1

Reviewer 1 Report

Comments and Suggestions for Authors

This paper examined the performance of European ESG leaders from 2012-2023. It expanded existing models (CAPM, FF3, FF5) by including VIX, return dispersion, and analyzing herding behavior. Results showed underperformance and negative alpha for ESG leaders, with a tilt towards specific styles. Herding behavior diminished diversification benefits. Including VIX, dispersion, and herding enhanced the analysis of ESG leaders' performance.

Literature review

·         The literatures which introduce volatility index (VIX) and dispersion (if there is any) into Fama-French models should be discussed in this paper.

     For example:

1.      Chen, X., & Gao, N. R. W. (2020). Revisiting Fama–French’s asset pricing model with an MCB volatility risk factor. The Journal of Risk Finance21(3), 233-251. 

            This paper introduced the term structure of VIX to Fama–French’s asset pricing model.

2.      Durand, R. B., Lim, D., & Zumwalt, J. K. (2011). Fear and the Fama‐French factors. Financial Management40(2), 409-426.

           In this paper, Fama and French’s three-factor model is augmented with a momentum   factor and the VIX.

Methodology

·         Authors should show that requirements for the OLS model are fulfilled. Adding new dependent variables to FF5 model and using the ESG indices time series data require tests being conducted to assess the assumptions and validity of the OLS model, such as multicollinearity test, normality test, etc. Test results should be discussed.

·         There is a duplicated “+” in rows 368 and 414.

Data

·         In row 242, for European ESG Leaders portfolios, the Euro zone risk free interest rate should be applied rather than the one-year Treasury Bill rate. Please refer to the link:

https://www.esma.europa.eu/benchmark-administrators/working-group-euro-risk-free-rates

Similarly for Global portfolio, a suitable rate which can represent a globally risk-free rate is better to be applied.

·         Please provide reasons for choosing the data period from 01/06/2012 to 15/07/2023.

·         Could you explain the rationale behind choosing the Sustainalytics ESG scores? Given the numerous ESG scores available in the financial industry, including internally developed scores by banks and funds, as well as those provided by third-party vendors such as Bloomberg, MSCI, Morningstar, and others, it would be helpful to understand the specific reasons for opting for Sustainalytics.

·         In the analysis of the EU ESG leaders' performance in FF7, it is recommended to utilize the EU Volatility Index (VSTOXX Index) instead of the US Volatility Index - VIX index.

o   The EU Volatility Index specifically measures the anticipated volatility in the Euro Stoxx 50 index over the next 30 days.

·         Similarly, for the regression test involving Global ESG leaders, it is recommended to incorporate an index representing global equity implied volatility.

o   Considering a combination of volatility indices from the US, EU, and APAC stocks, or exploring alternative methods, may be justified in this context. 

Results and Discussion

·         Please review the descriptions of the indices in Table 1 and 2 as they appear to be somewhat inconsistent with the actual indices listed. For instance, STOXX Global 1800 is described in Table 1 but not included in the table itself. Similarly, STOXX Nordic is mentioned in Table 1 but not utilized in the research. Furthermore, while the indices of the Leaders family are listed in Table 1 and 2, there is no corresponding explanation provided in Table 1.

·         In Table 5 and 6, and in rows 328 to 330, the ADF tests indicate that the financial indices exhibit stationarity. This finding appears to contradict the well-established understanding that financial time series data are typically non-stationary, as highlighted in the work of Lopez de Prado (2018). One of the challenges of quantitative analysis in finance is that time series of prices have trends or a non-constant mean. This makes the time series non-stationary.

Lopez de Prado, M. (2018). Advances in financial machine learning Chapter 5. John Wiley & Sons.

·         In Table 7 and 9, betas are significantly smaller than 1. However, they are still classified as medium betas. This raises the question: why are they considered medium betas? According to https://www.suredividend.com/low-beta-stocks/, the 5 lowest beta stocks’ betas are much higher than the betas in Table 7.

·         In the regression models for benchmark indices such as the Stoxx Europe 600 Index and Stoxx Global 1800 Index, their betas should ideally be 1 since they represent the overall market. However, in Table 7 and 9, their betas are significantly below 1. This raises the question as to why their betas deviate significantly from the expected value of 1, and which market returns are being utilized in the regression models.

·         Please clarify whether the R Square in Table 7 and 9 are adjusted R Square. It is recommended to use the adjusted R Square.

·         Based on the results from Table 9 and 10, if Model II does not indicate herding behavior during up or down-market days, what does this imply? Does this suggest that herding behavior only exists in flat market days? Please provide further elaboration.

·         Could you please provide further clarification on Figure 3 and 4? Do the green bars represent daily returns of leaders’ indices? And how did these figures visualize the herding behaviors?

Comments on the Quality of English Language

The authors use appropriate academic language and terminology throughout the paper, demonstrating a good command of the subject matter. The sentences are generally well-formed, and the ideas are expressed clearly. While the overall language quality is good, a moderate English editing could help enhance clarity and readability of this paper.

Reviewer 2 Report

Comments and Suggestions for Authors

This paper examines the performance of ESG strategies in Europe finding negative alpha associated with ESG leaders. The paper is well-written, the methodology appropriate and the conclusions interesting.

One minor suggestion is to consider rewording the description of the existing literature as "poor." Perhaps an alternative phrasing such as "underdeveloped" would work, but even then, there has been much work on ESG and investing as is well-documented in the paper's quality literature review. I understand the interest in motivating the contribution, but I think it is possible without the approach to oversell the lack of research in this area.

Round 2

Reviewer 1 Report

Comments and Suggestions for Authors

The authors have effectively incorporated the necessary revisions as suggested. Substantial changes have been made to the data, methodology and results. Additionally, expanded analyses and discussions are provided as suggested, which have further strengthened the robustness of the findings.

The revised version now demonstrates a clear and concise presentation of the research findings, and the modifications have significantly strengthened the overall quality of the paper.

Comments on the Quality of English Language

Overall, the quality of the English is good. However, minor editing could further enhance clarity and ensure smooth readability.